# SARAL's Full Mission Reprocessing: Improvement with the GDR-F Standard

**Ghita Jettou** [1,*]**, Manon Rousseau** [1]**, Fanny Piras** [1] **, Mathilde Simeon** [2] **and Ngan Tran** [1]

1 Collecte Localisation Satellites, 31520 Ramonville, France
2 Centre National d'Etudes Spatiales, 31400 Toulouse, France
* Correspondence: gjettou@groupcls.com

**Abstract:** Seven years (2013–2019) of the French/Indian mission SARAL altimetry data have been successfully reprocessed within the SALP contract supported by CNES to produce a new data set of GDR (Geophysical Data Record) using an updated, modern set of algorithms and models. The main objective of this article is to assess the quality of the reprocessed dataset and estimate the system's performance using GDR-F products. To achieve this goal, the new dataset has been validated against the previous one (identified as GDR-T) using mono-mission metrics and comparisons to reference altimetry missions such as Jason-2 and Jason-3. The new data set shows a clear improvement in data quality. The product validation shows a reduction of the standard deviation of crossovers' Sea Surface Height differences from 5.5 cm (GDR-T) to 5.2 cm (GDR-F). This paper presents the main processing changes and shows some of the results from the validation and quality-assurance processes. The major improvement of the GDR-F data set with respect to the previous one is due to the use of state-of-the-art orbit standards (POE-F) and geophysical corrections, including new tidal models, a new wet troposphere retrieval algorithm, and a new algorithm for sea state estimation. The intent of this paper is to highlight the overall benefit of this new dataset.

**Keywords:** SARAL; AltiKa; altimetry; reprocessing; GDR-F altimetry standard



## 1. Introduction

The French/Indian mission SARAL was launched on 25 February 2013, into a polar orbit with an inclination of 98.55°. SARAL's platform carries the first Ka-band radar altimeter (AltiKa), with its variable pulse repetition frequency and reduced footprint, and a dual-frequency (23.8 GHz/37 GHz) nadir-pointing radiometer. The key purpose of the microwave radiometer is to provide an accurate tropospheric correction to the range measurements retrieved by the altimeter. An overview of the mission is given in [1].

Scientists making use of altimetry data often need a long time series of data to be able to accurately characterize trends and cycles for geophysical parameters. Datasets such as these can only be compiled by consolidating and homogenizing observations made by several missions. To achieve this, the approach is to conduct a reprocessing activity designed to create a homogeneous altimetry product standard with a uniform set of algorithms and models, the so-called "GDR-F standard", where "GDR" stands for "Geophysical Data Record" and "F" refers to the F version of the orbit and the full set of revised geophysical corrections (detailed in Section 2.4). SARAL's dataset was the first one to be reprocessed with this new standard and was closely followed by Jason-3's dataset reprocessing (2021). Jason-2's GDR-F reprocessing is planned for 2023, and Jason-1's will follow. The reprocessing activity for SARAL's dataset has now concluded, and a quality assurance and scientific validation process has been performed on the output Level 2 products (L2-GDR).

Since the launch, SARAL's L2 Geophysical Data Records version T (GDR-T) have been monitored to assess the quality of the mission's data. Quality reports summarizing mission

performances are generated on a cyclic basis and made available through the AVISO web page [2]. Main calibration/validation activities and metrics are summarized in yearly reports, available at [3]. Continuous monitoring represents a solid basis for comparison for reprocessed datasets. The analysis hereafter is based on a full comparison using mono-mission metrics through the comparison of altimeter and radiometer parameters between both GDR-T and GDR-F standard L2 products. Multi-mission comparisons were also carried out, as they allow assessing mission biases, parameter discrepancies, Sea Surface Height (SSH) consistency, etc.

The reprocessed dataset covers the period from 14 March 2013 (beginning of cycle 1) to 11 November 2019 (end of cycle 134). Cycles 135 onward were originally produced with the new GDR-F standard. SARAL's dataset is provided on a pass-by-pass basis, where each file contains an ascending or descending orbital track going from pole to pole, and it can be obtained through the AVISO data service, as explained in https://www.aviso.altimetry.fr/en/data/products/sea-surface-height-products/global/gdr-igdr-and-ogdr.html (accessed on 7 May 2023). This paper presents the necessary background information to allow the user to fully understand the content and the overall benefit of the reprocessed dataset.

## 2. Materials and Methods

### 2.1. Orbit

Errors in the knowledge of the altitude translate directly into errors in the surface height. Hence, errors in knowledge of the rate of change of altitude also translate into surface height errors via the Doppler correction to range. Moreover, errors in the along-track positioning appear as errors in the measurement of the time tag. For these reasons, the use of an accurate orbit solution is an essential first step in providing an accurate dataset. To produce a high-quality orbit solution for the GDR-F products, the CNES (Centre Nationale d'Etudes Spatiales) POD (Precise Orbit Determination) Center computed a new precise orbit solution for SARAL's dataset, the so-called POE-F (Precise Orbit Ephemeris-version F) orbit solution. The details of the computation of this orbit solution are given in [4]. With this solution, radial errors were found to be reduced from 1.3 cm to 7.6 mm when compared to the POE-E reference orbit used in GDR-T.

### 2.2. Microwave Radiometer Derived Parameters

The altimeter is coupled with a Microwave Radiometer (MWR) to correct for the excess path delay induced by water vapor in the atmosphere. SARAL's MWR is a dual-frequency instrument providing measurements at 23.8 GHz and 37 GHz. The MWR Brightness Temperatures (BTs) dataset was reprocessed to correct for the Hot Count Saturation Anomaly (HCSA) that occurred between cycles 3 and 7. During this event, hot calibration counts reached the analogic/numeric converter voltage limit, leading to saturated count values. It was corrected on board on 22 October 2013, with the update of the offset in the database; see [5] for further details. In the GDR-F reprocessing, a new algorithm was implemented to correct for the saturation effect of HCSA (between cycles 3 and 7) on the gains and residual temperatures used to compute the brightness temperatures of the 37 GHz channel. The left plot of Figure 1 shows that with the GDR-F standard (red curve), the brightness temperature of the Ka band has a normal behavior during the saturation period (visible on the green curve). Hence, the parameters retrieved using the brightness temperatures, such as atmospheric attenuation (right plot of Figure 1), are no longer impacted by the saturation anomaly in the GDR-F version. The bias observed between the two versions of attenuation comes from changes in the neural network retrieval coefficients.

The GDR-F standard introduces a new algorithm to interpolate the radiometer measurements (5 Hz) at the altimeter's (1 Hz) time tag. To avoid land contamination near shore, radiometer measurements are selected according to their surface type before 1 Hz averaging. MWR measurements within a window of 1 s and on the same surface type as the altimeter are selected for the averaging. If no such measurements can be found, an extended window is used, and the closest value is selected. The corrected and interpolated BTs are

then used as an input to the L2 processing for the computation of the Wet Troposphere Correction (WTC), liquid water content, water vapor content, and atmospheric attenuation.

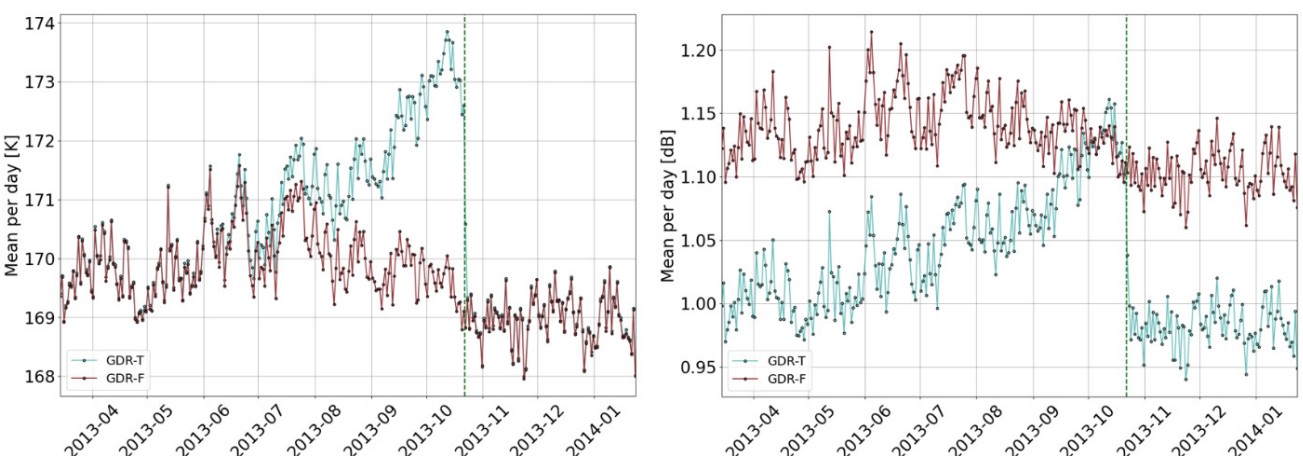

**Figure 1.** Daily mean of GDR-T (green) and GDR-F (red) parameters: (**left**) Ka-band brightness temperature and (**right**) atmospheric attenuation correction.

The L2 processing neural network learning database is built on simulations using a radiative transfer model. The two brightness temperatures and the altimeter backscattering, which provides surface roughness information, are used to retrieve radiometer-derived geophysical parameters. As performed for the ENVISAT v3.0 dataset, two additional input parameters are also included in the learning database of the new neural network [6]:

- The sea surface temperature (SST) for a better estimation of the surface emissivity. In the GDR-F products, the NOAA (National Oceanic and Atmospheric Administration) OISST (Optimum Interpolation SST) is used as input.
- The atmospheric temperature lapse rate ($\gamma$ 800), the temperature decrease slope from the surface to the 800 hPa layer of the atmosphere, that reduces systematic biases over upwelling regions. A climatological map of $\gamma$ 800 is used as input.

The retrieval of WTC for low-wind situations was also improved with respect to the GDR-T standard since the radiative transfer model used for the simulations was improved. This new physical approach shows a level of performance close to Jason missions (see dedicated Section 3.2.4). The same input algorithm is used to estimate the total column water. As for the cloud liquid water content, only the 3-input algorithm is used.

### 2.3. Level 2 Processing

The "retracking algorithm", or "retracker", is the on-ground processing step that consists of retrieving geophysical variables by inversion of the backscattered echo. In the L2 reprocessing chain, the same four retrackers used for the GDR-T dataset are executed for all records. Five retrackers are made available in the GDR-F products (see [4] for further details), but only the "ocean retracker" is assessed in this study. The so-called ocean retracker is the MLE4 retracking algorithm [7] that fits the Brown analytical model (as described in [8]) to the backscattered echo, allowing the retrieval of the following parameters:

- Altimeter range;
- SWH (significant Wave Height);
- Sigma naught;
- Square of the mispointing angle.

In addition to range, Sigma naught, significant wave height, and mispointing angle (20 Hz) estimates, a 1-Hz-compressed value is produced for these four main parameters.

One limitation of the MLE4 retracker is the need for look-up tables to compensate for the error made by modeling the point target response by a Gaussian function [9]. In the case of the Jason missions, look-up tables have been implemented in the ground segment for the range, the SWH, and the Sigma naught parameters, with the correction on the mispointing parameter being considered negligible [10]. A similar approach was initially chosen for SARAL and implemented in the GDR-T version.

However, a SARAL specificity must be considered in the look-up tables: its antenna aperture is smaller than the Jason missions ($0.65°$ versus $1.29°$) and implies a non-negligible impact of the gaussian approximation of the antenna diagram in the Brown model [11]. The figures hereafter show the difference between the real antenna diagram values and the Gaussian approximation (as used in the Brown model) for AltiKa (SARAL's altimeter) on Figure 2 and Poseidon-3b (Jason-3's altimeter) on Figure 3.

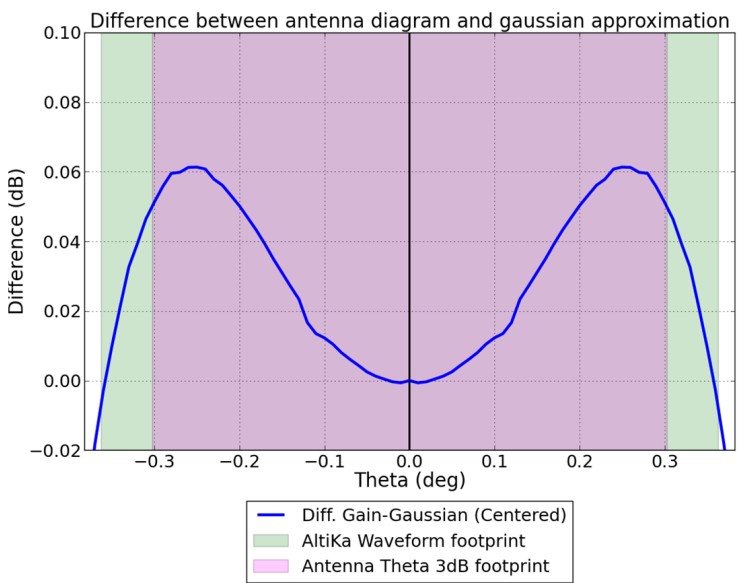

**Figure 2.** Difference between Gaussian approximation and real antenna diagram for AltiKa (courtesy of S. Le Gac).

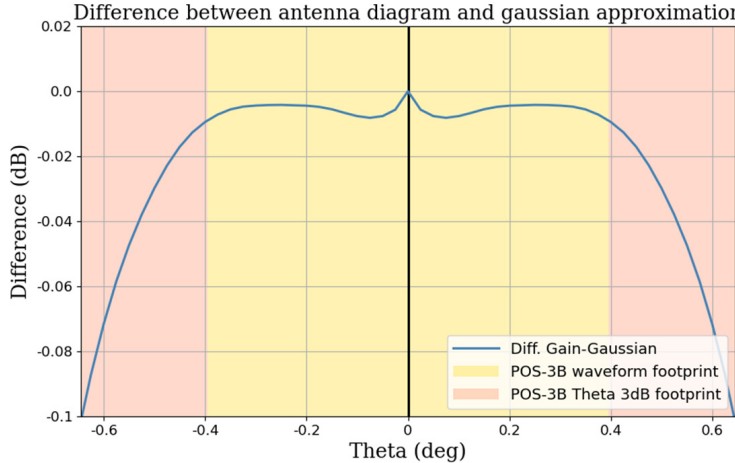

**Figure 3.** Difference between Gaussian approximation and real antenna diagram for Poseidon-3B.

Firstly, one can observe that the AltiKa antenna diagram reaches its $-3$ dB within the waveform footprint, which is not the case for Poseidon-3B, which has a wider antenna aperture. Secondly, the difference between the real antenna diagram and the Gaussian

approximation reaches more than 0.06 dB within the waveform footprint for AltiKa, versus 0.01 dB for Poseidon-3B. As observed in Figure 4, the Gaussian approximation of the antenna diagram for AltiKa has a non-negligible impact on the MLE4 estimates.

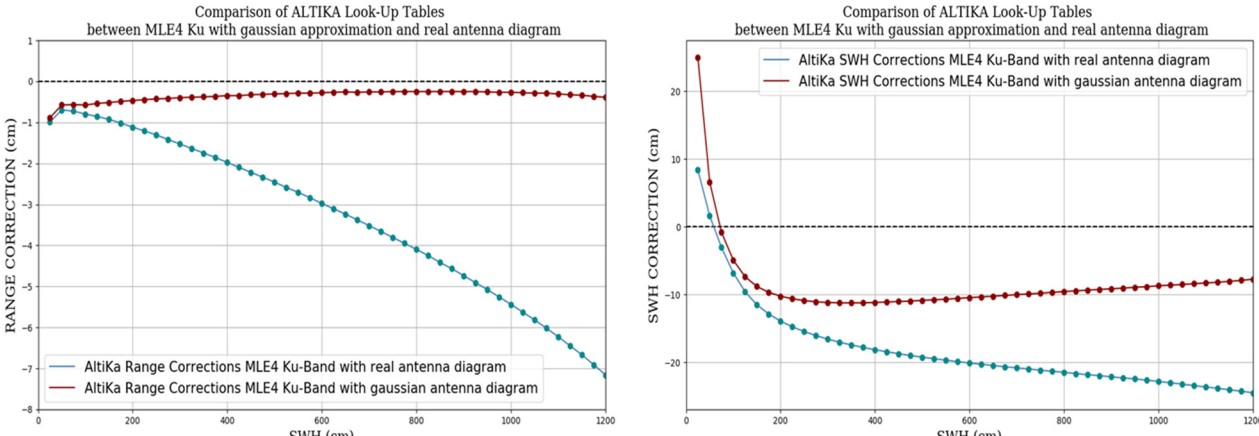

**Figure 4.** LUTs for (**left**) range and (**right**) SWH. The red curves represent GDR-T LUTs considering only the real PTR, and the blue curves represent GDR-F LUTs taking into account the real AltiKa antenna diagram.

New Look-Up Tables (LUTs) have been computed with the simulator using, in addition to the real Point Target Response, the real antenna diagram. The result is simulated echoes using a double convolution instead of a simple convolution when considering GDR-T LUTs. Figure 4 shows the difference between the previous GDR-T look-up tables for range and SWH, with the real PTR and a Gaussian approximation of the antenna diagram (red curve), and the new GDR-F look-up tables, with the real PTR and the real AltiKa antenna diagram values (blue curve). Range and SWH are more sensitive to wave height when using the real antenna diagram. Range differences can reach up to 7 cm, and SWH differences vary from 0 cm to 12 cm, both depending on the significant wave height values.

This effect also impacts the mispointing parameter, for which the LUTs are no longer negligible, as shown in Figure 5, and therefore must be applied to correct the mispointing parameter. The global impact of this new look-up table applied to the L2 products is described in Section 3.2.1.

Note that the LUTs for Sigma naught are negligible and therefore not detailed in this paper.

### 2.4. Auxiliary Models

Many geophysical and meteorological auxiliary models are used in the processing of altimetry data. Establishing a common baseline of models to be used in the processing of data sets from different missions is helpful when trying to consolidate data (Table 1). The creation of more accurate models is continually the topic of on-going research, and the models that are now available are an improvement to those used during the original processing of GDR products. A full set of meteorological and geophysical corrections are provided in the L2 GDR-F product for the user to apply to the range values (see [4] for further details):

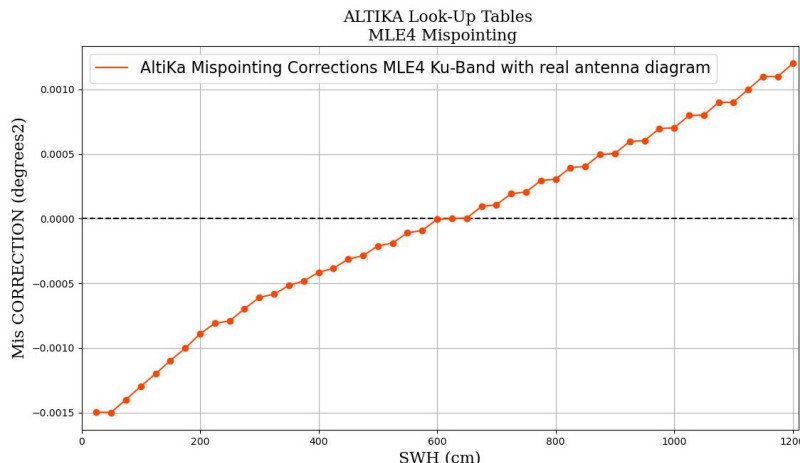

**Figure 5.** GDR-F MLE4 LUT for the mispointing parameter (square off-nadir angle), where the real PTR and the real antenna diagram value were considered.

**Table 1.** GDR-T and GDR-F geophysical models.

| MODEL | GDR-T | GDR-F |
|---|---|---|
| **DRY TROPOSPHERE RANGE CORRECTION** | Using ECMWF atmospheric pressures and models for S1 and S2 atmospheric tides | |
| **WET TROPOSPHERE RANGE CORRECTION FROM MODEL** | ECMWF model | |
| **IONOSPHERE CORRECTION** | Global Ionosphere TEC Maps from JPL | |
| **SEA STATE BIAS** | Hybrid SSB model from [12] | Two empirical solutions were fitted to one year of SARAL GDR-F data:<br>- a 2-parameter model using SWH and wind speed from altimeter measurements (standard version)<br>- a 3-parameter model using SWH, wind speed, and the mean wave period T02 from the Meteo-France WAM model [13] |
| **BATHYMETRY** | DTM2000.1 | ACE2 (from EAPRS Laboratory) |
| **MEAN SEA SURFACE MODEL** | MSS CNES-CLS11, MSS CNES CLS15 from cycle 34 | CNES-CLS 2015 model [14] and DTU 2015 model [15] |
| **MEAN DYNAMIC TOPOGRAPHY MODEL** | MDT CNES-CLS09 | MDT_CNES-CLS-2018 [16] |
| **GEOID** | MDT CNES-CLS09 | EGM2008 |
| **INVERSE BAROMETER CORRECTION** | Using ECMWF atmospheric pressures after removing S1 and S2 atmospheric tides | |
| **NON-TIDAL HIGH FREQUENCY DEALIASING CORRECTION** | Mog2D high-resolution ocean model [17] | |
| **OCEAN TIDE** | GOT4.8<br>FES2012 + S1 and M4 ocean tides<br>S1 and M4 load tides are ignored | GOT4.10c model and FES2014b |
| **SOLID EARTH TIDE** | From tidal potential of Cartwright and Edden [1973] Corrected tables of tidal harmonics [18] | |
| **POLE TIDE** | Equilibrium model | Desai model [19] with updated Mean Pole Location (MPL) [20] |
| **INTERNAL TIDE** | None | HRET-v7.0 model [21] |
| **WIND SPEED MODEL** | ECMWF model | ECMWF model |

For Sea Surface Height (SSH) values, the appropriate set of corrections, as detailed in [4], chosen based on availability and surface type, are applied during the L2 processing. The appropriate set of corrections for land are the dry and wet tropospheric corrections, the ionospheric correction, the solid-earth and pole tides, and the ocean-loading component (only) of the ocean tide. Over the ocean, the inverse barometric correction and the remainder of the ocean tides are also accounted for.

## 3. Results and Discussions

### 3.1. Validation Overview

Validation of GDR-F products was performed using an initial dataset of one year's worth of SARAL's dataset (year 2015). Once the validation process was complete, this preliminary dataset was used to fine-tune the wind speed retrieval algorithm by adapting the Sigma naught bias to the new Sigma naught distribution and to derive the appropriate empirical Sea State Bias (SSB) corrections for the updated SSH computation. The entire data set was then reprocessed with the updated set of algorithms. The results presented in the following sections are based on the analysis of almost seven years, spanning from 14 March 2013, to 11 November 2019, which corresponds to cycles 1 to 134. Cycle 135 onwards were produced under the "F" standard and validated as part of operational Cal/Val activities. The data quality and performance of the reprocessed dataset were compared to the original GDR-T dataset and to the GDR-E version of Jason-2 and Jason-3 data, which were updated with consistent geophysical corrections and standards when possible. Note that the results shown in this work were produced before the availability of Jason-3's GDR-F dataset.

In the following sections, usual editing procedures based on thresholds are used, and the editing strategy is dedicated to the deep open ocean. The thresholds applied to select valid measurements can be found in [4]. When comparisons are made with Jason missions, a specific data selection is applied to limit comparisons to the same geographical coverage. This selection is detailed in Section 3.3.1.

### 3.2. Analysis of Altimeter and Radiometer Parameters

#### 3.2.1. Waveform-Derived Square Off-Nadir Angle

The square off-nadir angle is one of the four parameters estimated by the MLE4 retracking algorithm. As explained in Section 2.3, the Gaussian approximation of the antenna diagram implies non-negligible biases in the retrieved geophysical parameters. The new LUTs introduced by the GDR-F standard and shown in Figure 5 drastically reduce the mispointing's wave dependency, as shown in Figures 6 and 7. Please note that the period used for the figures below was shortened to December 2018 because of the Star Sensor Anomaly (SSA) that affects the nadir pointing accuracy from February 2019 onwards (further details about the SSA are available in the 2020 Annual Performance Report [3]).

#### 3.2.2. Backscatter Coefficient and Wind Speed

The backscatter coefficient parameter $\sigma 0$ is estimated by the retracking of the echoes. The Sigma naught is linked, by definition, to the amplitude of the waveform [8].

Among the four parameters estimated by the MLE4 algorithm, the Sigma naught $\sigma 0$ is the only one not affected by the introduction of the real antenna diagram; the new look-up table correction is still negligible for this parameter, and therefore no $\sigma 0$ LUTs are considered in the GDR-F L2 products. However, a new normalizing strategy for the Low-Pass Filter (LPF) has been adopted: it is now normalized by its averaged value and not its maximum value. This changes the global amplitude of the echoes and thus changes the Sigma naught mean values, as shown in Figure 8, where a positive bias of 0.1 dB is now observed.

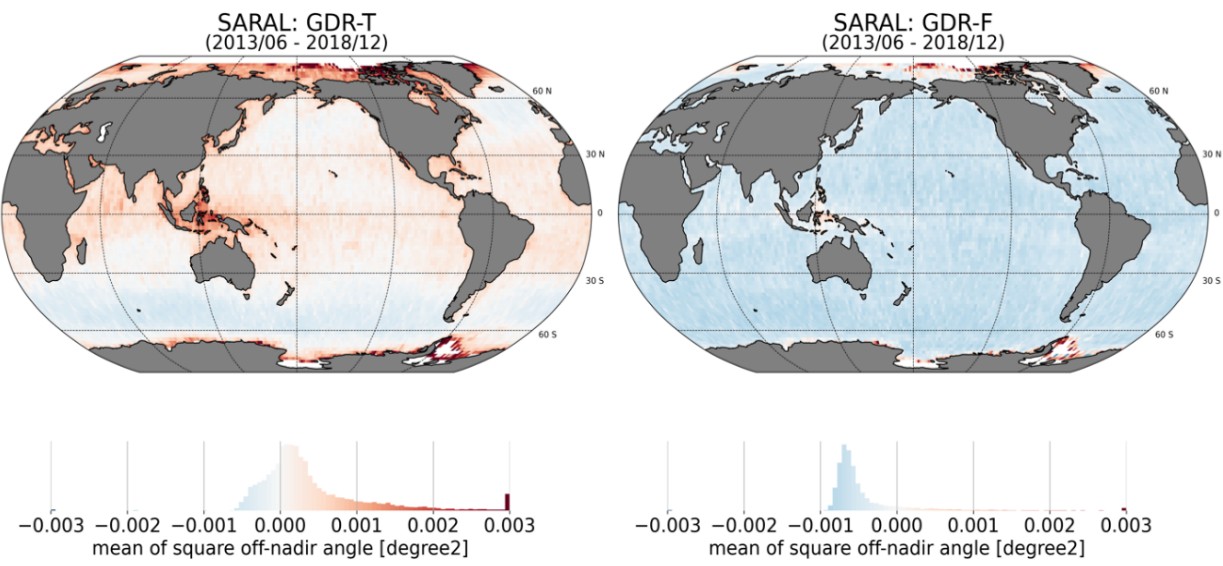

**Figure 6.** Global maps of waveform-derived mispointing (square off-nadir angle): (**left**) with the GDR-T dataset and (**right**) with the GDR-F dataset.

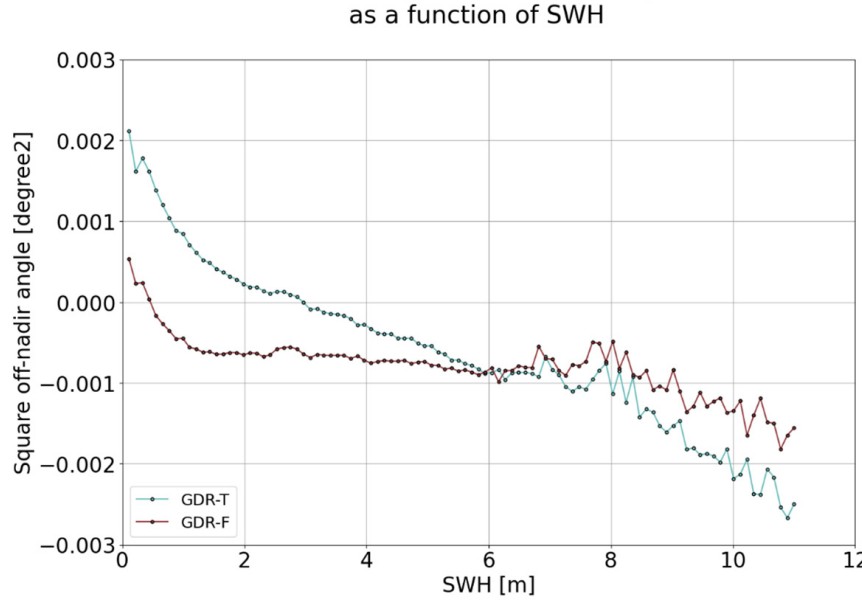

**Figure 7.** Square off-nadir angle (mispointing) as a function of SWH. For average waves (2–6 m), the FDR-F version shows no dependency compared to GDR-T with respect to SWH.

In the frame of the CNES PEACHI project [22], a two-parameter wind speed retrieval algorithm was developed for SARAL's data, similar to the one used for the Jason missions [23], which is based on both Sigma naught and SWH. This 2D version was tuned using 1 year of SARAL GDR-T/ASCAT-A scatterometer-collocated data [24]. The ASCAT-A products correspond to the operational NRT level 2 products with a 12.5 km sampling processed by KNMI/OSI-SAF (doi:10.15770/EUM_SAF_OSI_0007). This new wind speed model was validated against Jason-2 altimeter estimates and buoy data [24] and shows better agreement with the references than the 1D model [25] used to generate the GDR-T products in both comparison cases. Due to changes between GDR-T and GDR-F Sigma naught, a bias was determined and applied within the wind speed computation to correctly use the 2D model.

The comparison between the new GDR-F wind speed estimations and Jason-3 data at crossover locations shown in Figure 9 confirms the good performance of the algorithm. The relationship between the two missions' wind speeds is now truly linear, and SARAL's wind speed no longer saturates for values greater than 16 m/s, which tended to underestimate the wind speed compared to Jason-3.

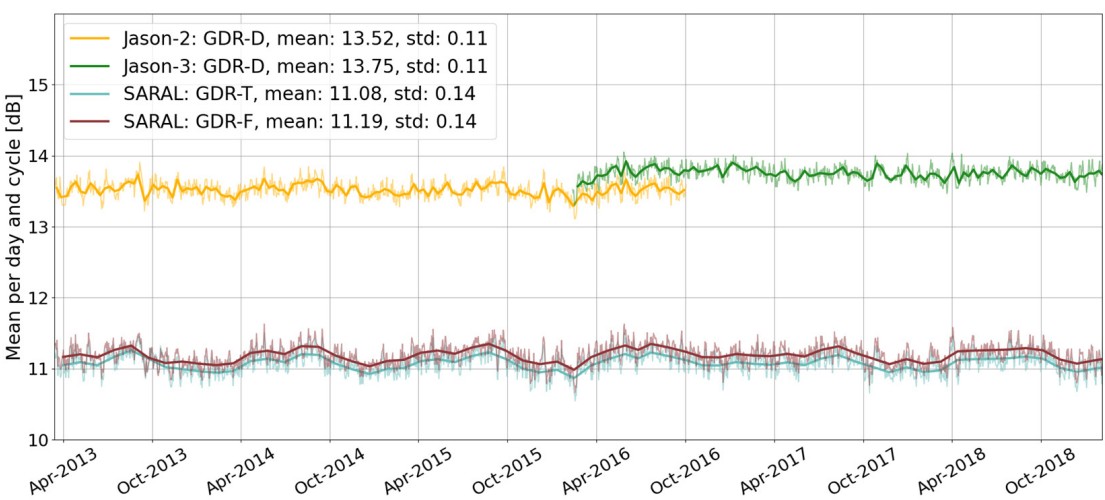

**Figure 8.** Daily (light lines) and cyclic (dark lines) monitoring of mean Sigma naught. Please note that Jasons have a 10-day cycle, whereas SARAL has a 35-day cycle.

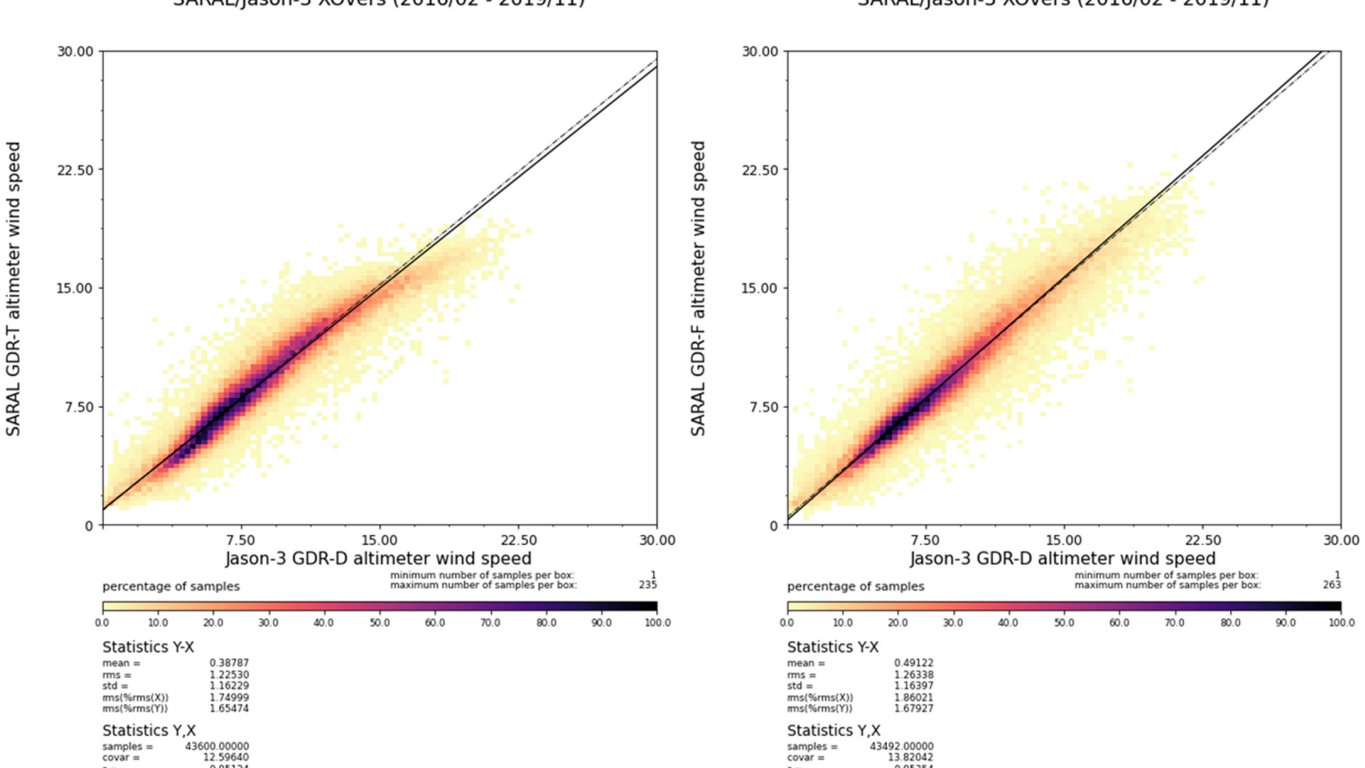

**Figure 9.** Dispersion diagram of wind speed between SARAL/AltiKa and Jason-3 at 3 h crossover points, using GDR-T standard (**left**) and GDR-F (**right**).

### 3.2.3. Significant Wave Height (SWH) and Sea State Bias (SSB)

The Significant Wave Height (SWH) is derived from the Sigma C [8], estimated by the retracking of the echoes:

$$SWH = 2c\sqrt{\sigma_c{}^2 - \sigma_p{}^2} \tag{1}$$

where $\sigma_c$ is the speed of light, and $\sigma_p = 0.513T$ where T is the radar pulse duration. As explained in Section 2.3, look-up tables have been updated with the implementation of the real AltiKa antenna diagram. Figure 3 shows the difference between the GDR-T and GDR-F SWH estimates. The new LUTs impact the SWH by ~−5 cm for a 2-m wave and by ~−10 cm for a 6-m wave. This is consistent with the analysis on real data, where on average and for all sea states, a mean bias of −5 cm is observed between GDR-F and GDR-T (Figure 10).

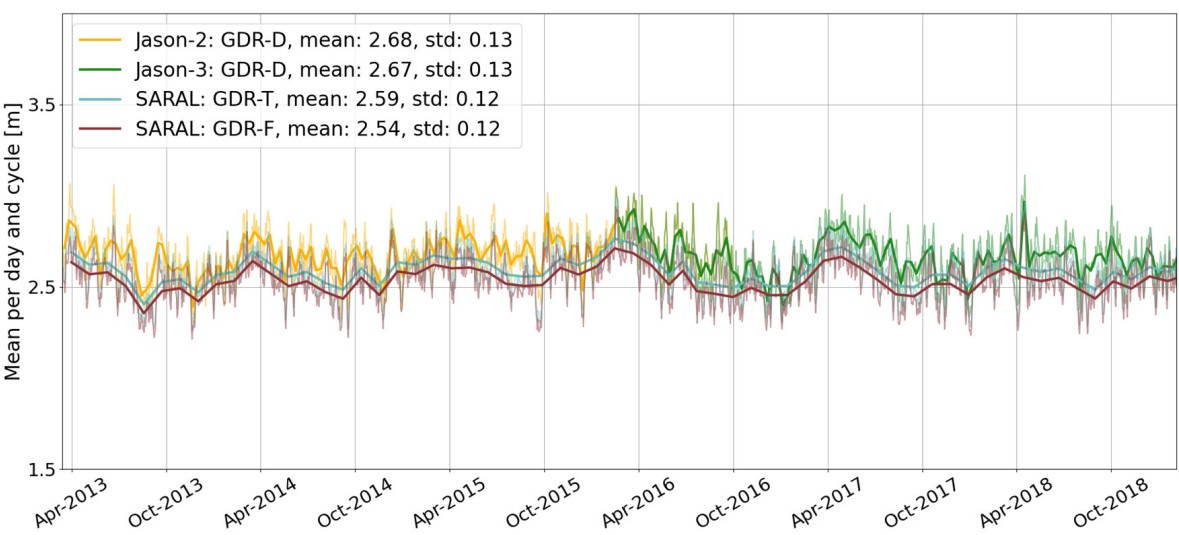

**Figure 10.** Daily (light lines) and cyclic (dark lines) monitoring of the mean SWH. Please note that Jason missions have a 10-day cycle, whereas SARAL missions have a 35-day cycle.

During the implementation of the GDR-F reprocessing chain, two new empirical SSB correction models have been derived based on the preliminary GDR-F-generated dataset. The first version relies on altimeter-derived SWH and WS (referred to as standard 2D), and the second one is based on the triplet: SWH, WS, and T02 (mean wave period) provided by the Météo France Wave Model (MFWAM) [26] (referred to as a 3D model in the L2 products). These models were developed with the latest version of the CLS non-parametric approach [27] and computed from crossover data.

Clear improvements are obtained with these new solutions, as seen in Figure 11.

On the left panel of Figure 11 we observe an average reduction of SSH variance of about ~1 cm when comparing GDR-F and GDR-T SSH variances at crossovers. On the right panel, the 3D version outperforms the updated 2D version with a reduction of the SSH variance at crossovers of −1.07 cm². Note that the improvement related to the inclusion of the mean wave period parameter as a third input to develop a 3D SSB model was initially reported in [28] and confirmed here.

Note that 3D SSB corrections are provided in all GDR-F versions of altimeter products. They are available for the Jason-3 and TOPEX GDR-F reprocessed datasets. They have been provided in Sentinel-6 products since the launch of the mission and will be available after the reprocessing campaigns of Jason-1 and -2.

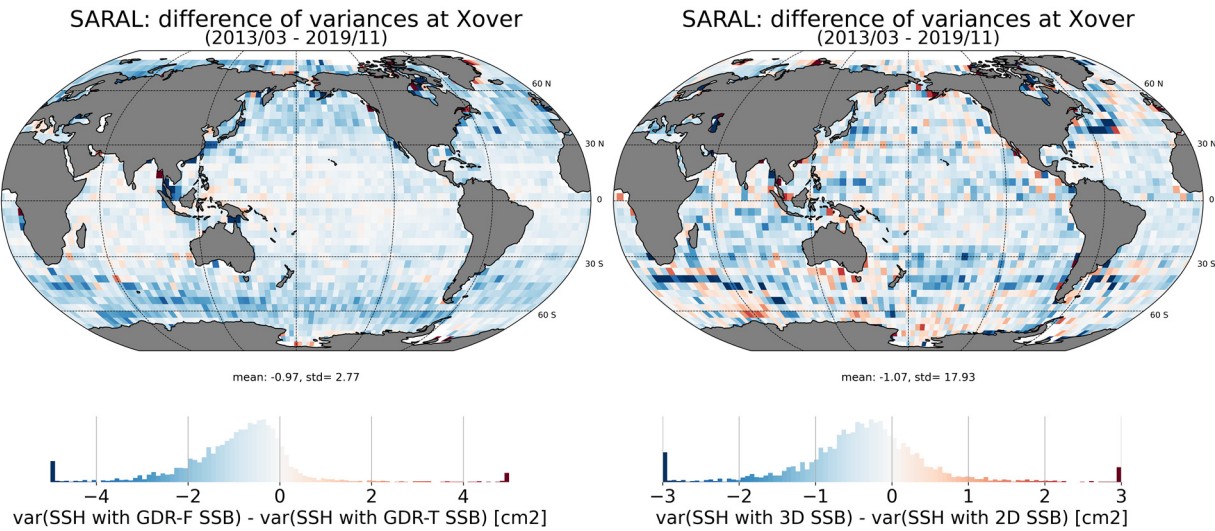

**Figure 11.** Maps of the difference in SSH variance at crossover points: (right) comparison of SSH variance at crossovers with 2D SSB models GDR-F-GDR-T comparisons of GDR-F SSH variance at crossovers using 2D-3D SSB models. The SSH variances are computed over $2° \times 2°$ boxes of crossovers. Negative values (blue in the maps) mean a variance reduction.

### 3.2.4. Wet Troposphere Correction (WTC)

As described in Section 2.2, GDR-F wet troposphere correction benefits from the introduction of a new 1 Hz interpolation algorithm and new neural network coefficients set for WTC retrieval.

Here, the ECWMF wet troposphere correction is used as a reference to investigate the differences between both versions of SARAL's datasets and Jason's radiometer corrections. GDR-T has globally dryer mean values of wet troposphere correction than GDR-F ones. The maps shown in Figure 12 (of radiometer and ECMWF model wet troposphere differences) are centered around the mean values (4.0 mm for GDR-T and −3.6 mm for GDR-F). The geographical structures on both maps are similar in high latitudes (around 50°N), but very different elsewhere.

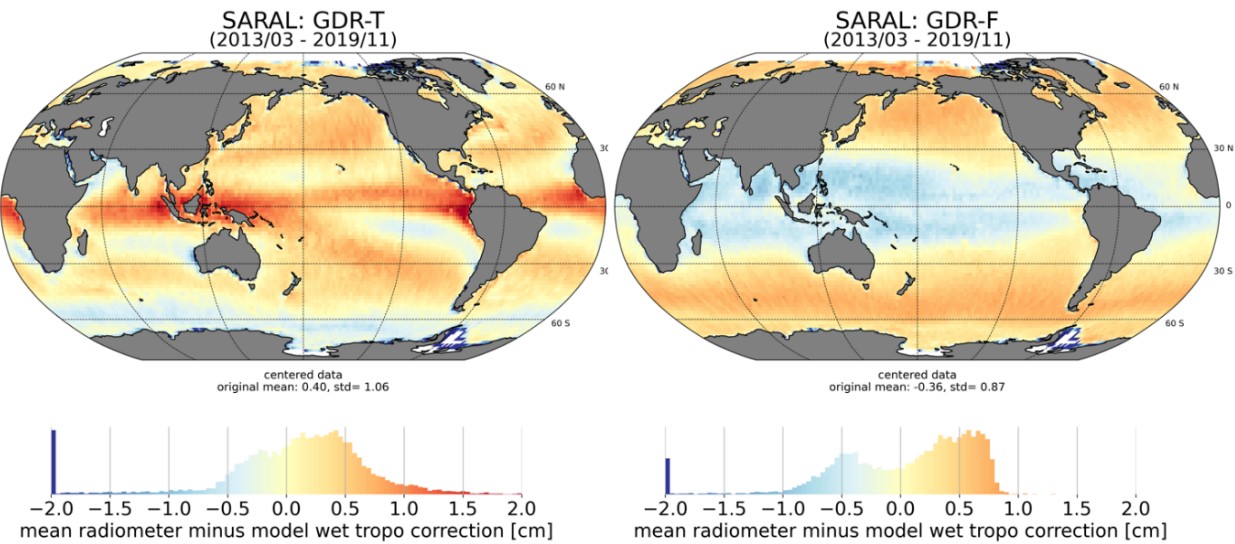

**Figure 12.** Maps of the radiometer wet troposphere correction difference with the model for GDR-T (**left**) and GDR-F (**right**). The high bias near the tropics and the precipitation signature are reduced with the GDR-F standard.

The information brought in by the new inputs (SST and the atmospheric lapse rate) in the neural network allows for the removal of the strong patterns in the tropic regions of high sea surface temperature and in the upwelling regions. In the southern high latitudes (below 50°S), it allows a better retrieval of areas of very low sea surface temperatures (compared to the model wet troposphere correction).

The standard deviation of the differences with the reference is lower for GDR-F (1.4 cm) compared to GDR-T (1.6 cm), as shown in Figure 13. Hence, the GDR-F version of the radiometer wet troposphere correction shows a closer performance with respect to the model compared to the GDR-T, slightly closer to the Jason missions.

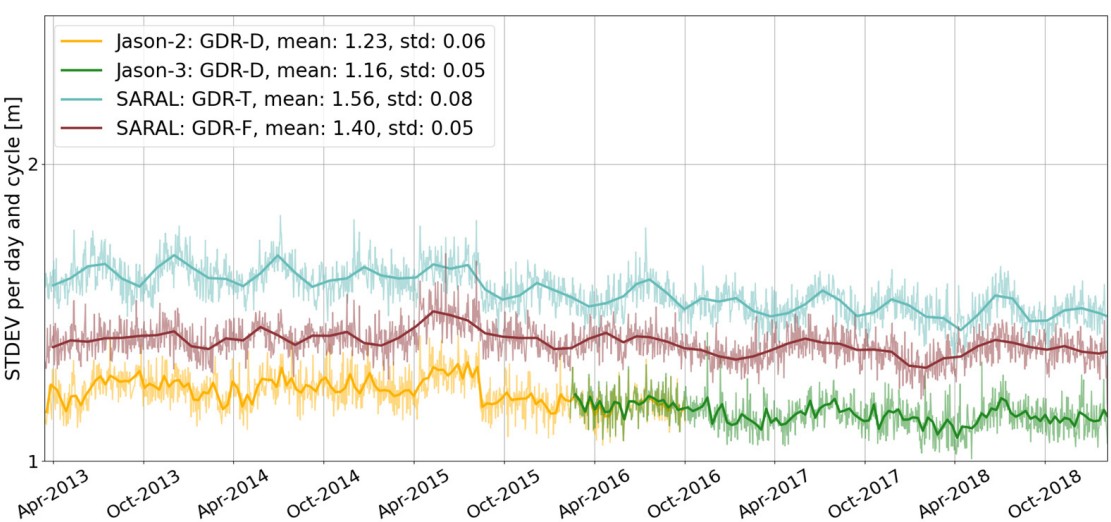

**Figure 13.** Daily (light lines) and cyclic (dark lines) monitoring of the standard deviation of wet tropospheric correction differences (radiometer minus model).

### 3.3. Sea Level Performance

3.3.1. Sea Surface Height (SSH) Cross-Over Analysis

SSH crossover differences are the main tool to analyze the altimetry system's performance. This gives a measure of the mission's performance on mesoscale time/space scales. The mean and standard deviation of SSH crossover differences are computed over valid datasets to characterize their geographical distribution and temporal evolution. When comparing two missions that cover different ground tracks, the geographical distribution of crossovers is not the same for both missions, which may bias the statistics. To overcome this issue and monitor the performances over stable surfaces and homogeneous distributions of crossovers, additional selections are applied:

- A maximum time difference of 10 days;
- High-latitude regions are not considered ($|\text{latitude}| < 66$);
- Weighting the crossover distribution by latitudes, where weighting depends on the crossovers' theoretical density. This also reduces the amplitude of the annual signal.

With this selection, SSH performances are estimated under equivalent conditions.

Figure 14 shows the maps of mean SSH differences at crossovers for the two versions of GDR products. Both standards show geographically correlated patterns of low amplitudes. Those patterns are not the same for GDR-T and GDR-F. Local differences can reach 2 cm with the GDR-T standard, while they remain below 1 cm for the GDR-F. In addition to mapping the differences, the global mean of SSH differences at crossovers is estimated for each cycle to assess systematic inconsistencies between ascending and descending tracks. Figure 15 shows a stable mean and standard deviation for both versions of data, with a weaker annual signal when using the GDR-F standard. This stability is also confirmed when

comparing Jason missions using the selection and weighting described earlier. The right panel of Figure 15 shows the monitoring of the cyclic standard deviation of SSH differences at crossovers. When using the GDR-F standard, SARAL shows a better performance than with the GDR-T standard, with an average standard deviation reduced to 5.2 cm compared to the previous 5.5 cm and a closer performance to Jason missions (around 4.9 cm). This performance is also illustrated in Figure 16, where we can see that the variance difference of SSH crossover differences is globally negative, indicating a decrease in variance and therefore a better performance.

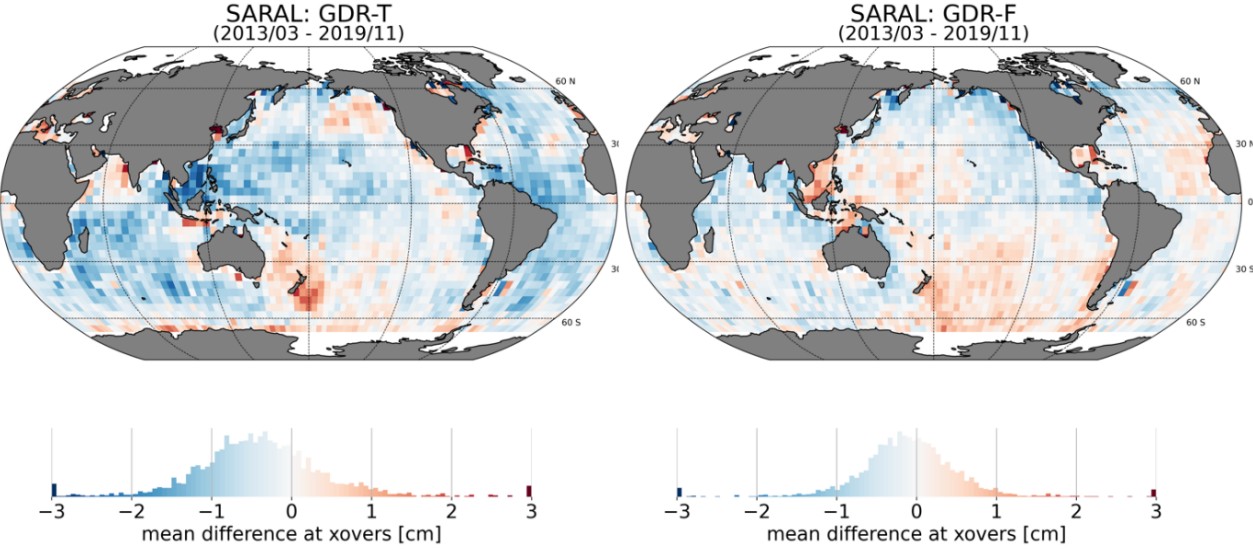

**Figure 14.** Maps of mean differences of SSH at crossover points: (**left**) using the GDR-T dataset; (**right**) using the GDR-F. $2° \times 2°$ box-averaged.

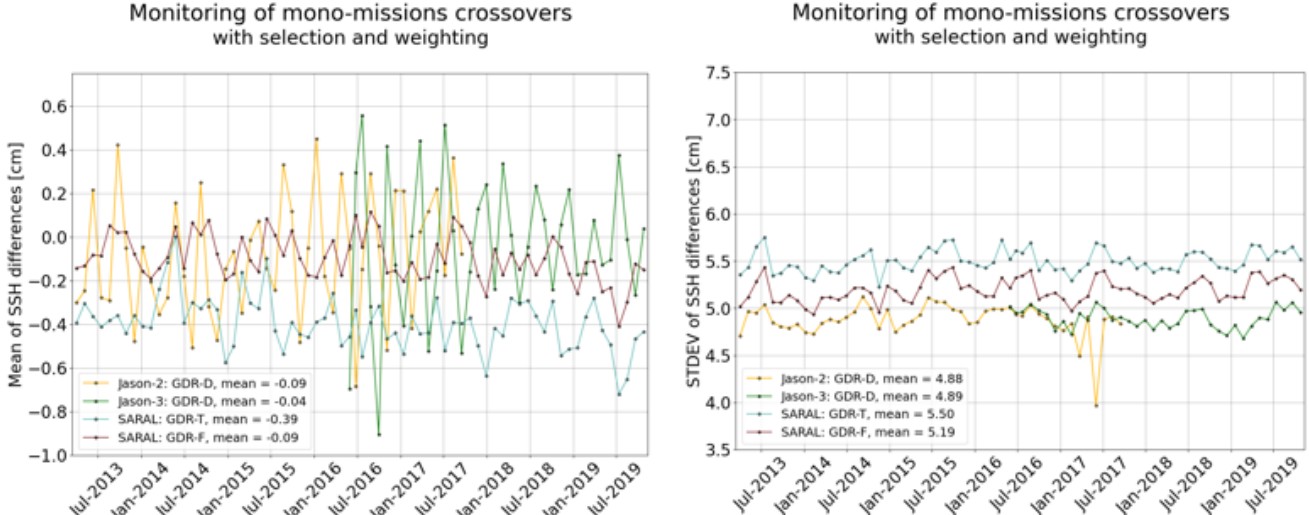

**Figure 15.** Cyclic monitoring of the mean (**left**) and standard deviation (**right**) of SSH differences at crossovers. The selection and weighting of crossover points are applied to have a comparable distribution of crossovers for SARAL and the Jason missions.

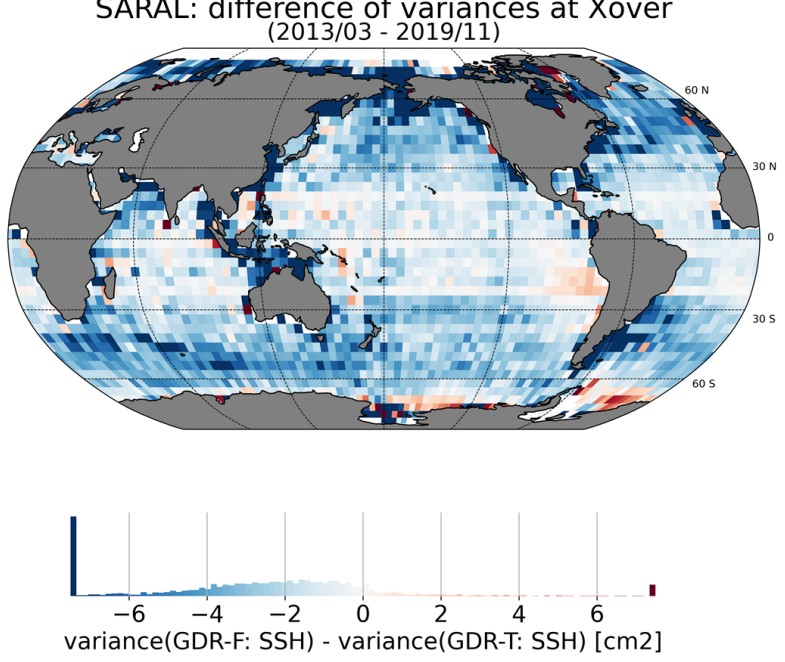

**Figure 16.** Maps of the difference of the variance of SSH differences at crossover: the variance of SSH differences is computed over $2° \times 2°$ boxes.

### 3.3.2. Analysis of Sea Level Anomalies (SLA) and Global Mean Sea Level (GMSL)

An on-track SLA provides additional metrics to estimate the system's performance. Figure 17 displays the temporal evolution of the mean and standard deviation of both versions of SARAL's data, Jason-2 and Jason-3 SLA. The evolution of the mean SLA allows the detection of shifts or drifts, whereas the variance can highlight changes in the long-term stability of the system. SARAL's GDR-F SLA shows similar signals and temporal evolution when compared to GDR-T and both Jason missions, with a bias of $-4$ cm when compared to Jason-3. The standard deviation of daily SLA is quite the same for both standards and Jason mission, around 10 cm.

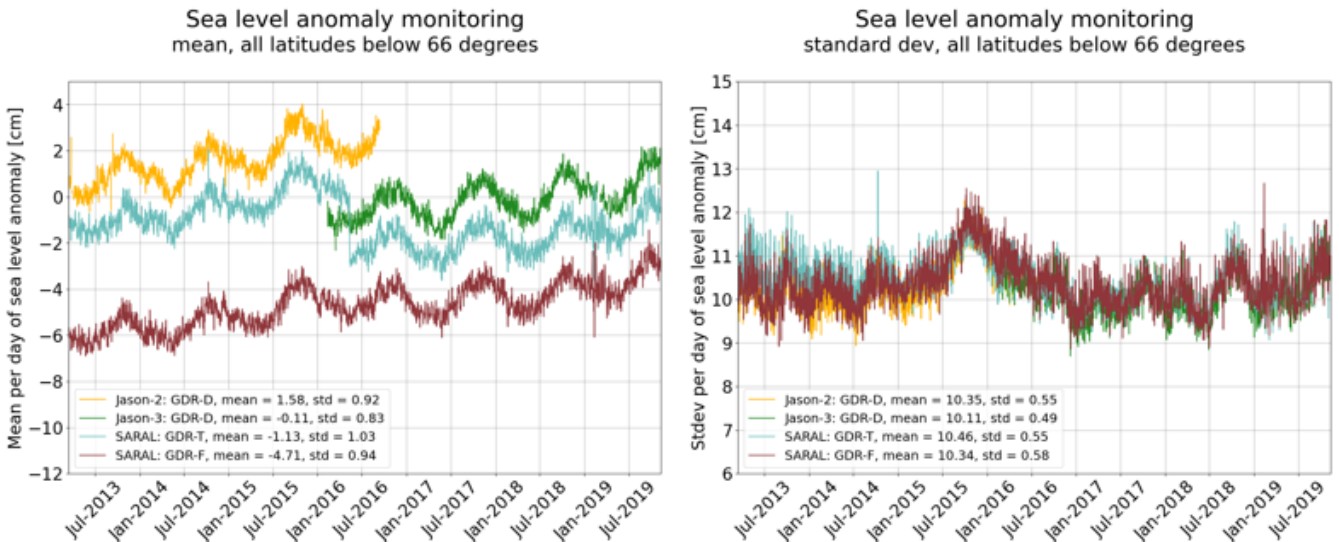

**Figure 17.** Daily monitoring of the mean (**left**) and standard deviation (**right**) of SLA. The selection of points with latitudes below $66°$ was applied to have comparable coverage for SARAL and the Jason missions.

Although GMSL monitoring is not one of the primary goals of the mission, Figure 18 displays how well SARAL's GMSL blends in with the reference GMSL record. Over the same period, GDR-F shows similar GMSL rise rates with a slightly closer trend to the reference series, with 3.6 mm/yr. for the GDR-F time series versus 4 mm/yr. when using the GDR-T standard.

**Figure 18.** SARAL/AltiKa global mean sea level record compared to the reference global mean sea level from TOPEX/Jason series. The time step for computing the GMSL is based on the reference missions' cyclic period of 10 days.

## 4. Conclusions

The reprocessed and operational GDR-F dataset presents ten years of SARAL altimetry data, validated and cross-calibrated both within the mission and in comparison with Jason-2/Jason-3. The data format is netCDF 4 to allow ease of access from a range of standard tools across the main computing platforms. The data set is fully described in the accompanying product handbook [4], and the self-documenting capabilities of netCDF have been used to present useful documentation within the data set itself. The GDR-F data set will therefore be useful and accessible to both researchers wishing to make use of the SARAL altimetry data and operational services using real-time data.

The assessment of the GDR-F data quality versus the GDR-T data shows a clear improvement in terms of accuracy and consistency with reference missions such as Jason-2 and Jason-3, with a reduction of the standard deviation of crossovers' Sea Surface Height differences from 5.5 cm (GDR-T) to 5.2 cm (GDR-F). The major improvements of the GDR-F dataset with respect to the previous GDR-T products come from the use of new reprocessed precise orbit solutions (POE-F), a new wet troposphere correction retrieval algorithm using corrected brightness temperatures from HCSA, a new sea-state bias, and state-of-the-art geophysical models. The comparison of the GDR-T and GDR-F error budgets detailed in [29] shows an overall reduction of Sea Surface Height error from 3.7 cm to 3.5 cm.

This assessment also identified some improvements for future reprocessing activities, such as the tuning of the trailing edge variation flag for the new mispointing distribution (considering the new LUTs as described in Section 3.2.1). Work on the next reprocessing of the SARAL altimetry data (GDR-G) is already planned. The work performed in the construction and operation of the GDR-F reprocessing chains has delivered an improved product and laid the groundwork for other reprocessing activities (Jason missions). The GDR-F dataset is a significant advance on the previously available GDR-T altimetry dataset, and the project team looks forward to feedback and results from the wider scientific community.

**Author Contributions:** Conceptualization, G.J.; methodology, G.J.; software, M.R.; validation, F.P., M.S. and N.T.; formal analysis, G.J., M.R. and F.P.; investigation, G.J., M.R., F.P. and M.S.; resources, G.J., M.R. and F.P.; data curation, M.R.; writing—original draft preparation, G.J.; writing—review and editing, G.J., M.R., F.P., M.S. and N.T.; visualization, G.J., M.R. and F.P. All authors have read and agreed to the published version of the manuscript.

**Funding:** This study was carried out in the framework of the SALP project, funded by the French spatial agency Centre National d'Etudes Spatiales (CNES).

**Data Availability Statement:** The access to the SARAL GDR-F dataset used in this work is described in https://www.aviso.altimetry.fr/en/data/products/sea-surface-height-products/global/gdr-igdr-and-ogdr.html (accessed on 7 May 2023).

**Acknowledgments:** I wish to acknowledge the help provided by the technical staff of the data processing team, M. Chauffaut, J. Dangreau, and F. Bailly-Poirot. I would like to extend my special thanks to F. Bignalet-Cazalet, A. Ollivier, M-I Pujol, and P. Prandi, for their valuable feedback that helped me finalize this paper.

**Conflicts of Interest:** The authors declare no conflict of interest. The funders had no role in the design of the study, in the collection, analysis, or interpretation of data, in the writing of the manuscript, or in the decision to publish the results.

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
