# Peer review of "SARAL’s Full Mission Reprocessing: Improvement with the GDR-F Standard"

_remotesensing, doi:10.3390/rs15102604_

Round 1

Reviewer 1 Report

This paper presents a timely and valuable study for SARAL's full mission reprocessing. The experiments were well designed and clearly documented. But I still have some minor comments left for authors to check and consider.
(1) The figure number are confusing. E.g., L248 Figure 3, L253 Figure 9, L262 Figue 10, L263 Figure 10(a), L293, Figure 10, etc.
(2) Figure 6, a legend or color bar is necessary to show specific range of square off-nadir angle.
(3) Figrue 10, the daily and cyclic monitoring of SWH was executed globally or regionally? If globally, within 66N and 66S? Why did Jason-derived SWHs always higher than SARAL?
(4) Figure 13, samilar comments for Figure 10 holds here.
(5) Figure 15, the mono-missions crossovers have some time-dependent regularity. But why does the peaks for GDR-F and troughs for GDR-T? If possible, please add some explanation.

Overall, I recommend this article to be published after a minor revision.

Reviewer 2 Report

The paper is attempt of showing the detailed analysis of GDR-F version of SARAL data. The topic is important and the paper is  informative about changes that happened in life time of SARAL. How ever there are many issues that has to be solved before accepting it for publication. Major observation listed below. 

1. At line 84  pl put the year after 22nd of October 

2. At this point author must explain how GDR-F algorithm handled the saturation of BT. Is count to BT conversion coefficients changed here. 

3. At line 105/106 mention sources of SST and lapse rate. 

4. At line 147 remove extra comma. 

5. At line 149/ 150 write SWH correction and range corrections and mention these are more sensitive when real antenna diagram is used. 

6. pl add color bar to figure -6 

7. kindly add a GDR -T vs Jason wind speed at crossover plot too and mention the standard stats like bias RMSE and correlation in figure -9 

8. At line 262, 'Clear improvements are obtained with these new solutions as seen in Figure 10. In Figure 10 (a) we observe an average reduction of SSH variance of about ~1 cm when comparing 263 GDR-F and GDR-T SSH variances at crossovers. In (b) side, the 3D version outperforms 264 the updated 2D version with a reduction of the SSH variance at crossovers of -1.07 cm². ' This entire para is about figure -11 and not 10 kindly revisit. Figure -11 also do not contain a color bar. 

9.  At line 248 'Figure 3 shows the difference between the GDR-T and GDR- 248 F SWH estimates'. It should be figure -10 

10. At line 292 'The standard deviation of the differences with the reference is lower for GDR-F (1.4 cm)  compared to GDR-T (1.6 cm) as shown Figure 10. Hence, GDR-F version of radiometer  wet troposphere correction shows a closer performance with respect to model compared  to GDR-T, slightly closer to Jason missions'. Again authors refer figure -10 (I think its their favorite) instead of figure -12. Here in explaining 12 it can written that WTC shows high bias with model near tropics and is having a precipitation signature in it which reduce in GRD-F WTC. 

Apart from these errors, a briefing about GDR-F and GDR-T processing would be good for general reader in introduction section. this should contain essential table of  information like algorithm and its details/ differences. SARAL has under gone geodetic mode in 2016 how these algorithm handles it. 

International cooperation related to SARAL is missing in this paper. 

Reviewer 3 Report

Review of SARAL’s Full Mission Reprocessing: improvement with the GDR-F standard

This paper provides a thorough assessment of the reprocessing of the SARAL radar altimetry mission to Geophysical Data Record Version F standards. As the only Ka-band nadir altimeter in the constellation of satellites, a published assessment of the performance of the latest processing chain is relevant for publication in Remote Sensing. Furthermore, SARAL was the first mission released in GDR-F standards, and an assessment of the improvements from the previous standard is valuable to the altimetry community.

The manuscript uses standard techniques in the altimetry community to assess instrument and geophysical corrections as well as the outputs from retracking the radar waveforms. There are no major technical issues with the analyses, and in that respect, the paper could be published with minor revisions. One major deficiency is that the comparisons to Jason-3 are to the previous processing standard, GDR-D. The full-mission reprocessing of Jason-3 has been available since January 2022, and a more valuable paper would compare GDR-F to GDR-F results. I would strongly urge the authors to consider add Jason-3 GDR-F results. In the current manuscript, the plots using Jason-2/3 data should labelled as GDR-C/D where appropriate (Fig. 8, 9, 10, 13,17, etc.)

In many places the manuscript would only be understood by an altimetry expert. In several cases, abbreviations are used without being defined (MWR, POD, etc.). (It would be better for GDR to be defined in line 42 when it is first introduced rather than in line 48.) The paper would be improved if a few sentences that summarize of the analyses in the paper were introduced either at the end of Section 1 or at the beginning of Section 2. At the moment, Section 2 begins abruptly with a series of subsections on each parameter with no context. 

The conclusions to the paper are very general. Would it be possible to add a table of an estimated error budgets for GDR-T and GDR-F? It is difficult to understand which improvements to the corrections have had the largest impact.

Comments:

Line 48: This sentence makes it sound like GDR-T files continue to be produced and monitored. Please reword it.

Line 81: An introductory sentence would be helpful to a general reader that explains the wet troposphere path delay and brightness temperatures.

Line 116: Again, another introductory sentence that better explains what retracking is would make this section clearer.

Figure 2: What are the black boxes?

Line 243: Please explain what sigma C is. Is there a citation for this equation?

Figure 10: Please note in the text or the caption that the cycle lengths are different for Jason versus SARAL.

Line 316: Please document specific details of the crossover analysis. What are the latitude ranges and shallow water and variability criteria. What is the maximum time difference for the crossovers?

Figure 15: Why was no crossover selection applied?

Figure 19: I’d recommend dropping the figure on the left. Were the trends calculated only over the 2013 to 2019 epoch? If so, they should be removed from the left figure. In the right figure, the trend would only be for J2/J3 and not TX/J1/J2/J3. 

Minor comments

Line 38: Dataset > Datasets

Line 74: CNES > the CNES

Line 103: “As done for” > “As done for the”

Line 125: comma needed after sigma naught

Line 144: gaussian > Gaussian

Line 147: delete extra comma

Figure 4 caption: diagramm > diagram

Line 229: similarly > similar

Line 230 [22 > [22]

Line 290: “it allows to better retrieve” > “it allows for better retrieval”

Line 295: to Jason missions > to the Jason missions

Reviewer 4 Report

General comments

The content of the paper is generally good. However, there is an issue with the paragraphing; some of them should be merged into one.

There is also an issue with the figure citations. For instance, Figures 11 ~ 13 are not cited in the text.

The picture quality (resolution or dpi) of the figures is quite low.

Also, there are some minor grammatical errors which the authors need to correct.

Specifics

Some punctuations are needed at line 125. Also, change sigma naught to Sigma0.

Lines 144 ~ 145 should not be a paragraph.

Check line 147.

Line 183, change one years’ to one year’s.

Line 203, change Waveform derived to Waveform-derived.

You should include colorbars to the maps in Figure 6.

Check lines 217 and 218, you must be consistent throughout. Which one you are using, is it Sigma0, Sigma Naught or σ0??

Line 229, change similarly to similar.

Line 230, check the citation [22].

Lines 252 ~ 253 should not be a paragraph.

Line 264, change (b) side to Figure 10b. Wait, wait, wait!!!! Figure 10 has no subplots. I think you are referring to Figure 11.

Check the grammar of the sentence in lines 284 ~ 286.

Line 289, change down to below.

Line 320 should not be a paragraph.

There should be a comma (,) after “Locally” in line 323.

How do you show Figure 17 before Figure 16?

Reviewer 5 Report

The work presented in this article seems to be fine, the graphs show that using the GDR-F method yields better results than with GDR-T, among other things. However, I consider that several changes should be made in the text to improve the text, so I consider that the following changes should be made

1.- Both the abstract and the introduction talk about two methods that are contrasted in the text, RDG-T and RDG-F, which in itself seems to be the central basis of the work. However, neither in the introduction nor in the methods is there a detailed description of the differences between the two methods. Therefore, I suggest that both the introduction and the methods provide a detailed description of the differences and advantages of each method.

2.- In line 94 you say: The GDR-F standard introduces a new algorithm to interpolate, however, you do not put what kind of interpolation it is, it seems correct to me that you describe this kind of things that sometimes are simple and obvious for you but not for the reader.

3. in section 3.2.4 it seems to me that you should indicate figures 11,12, and 13, if you are going to put them, it seems to me correct to include them in the text making a greater description than only in the figure caption.

4. In several figures you put dots and lines, but there are no dots so it is confusing to understand what you want to see, please fix.

5. The conclusion is quite vague, I think you should put more emphasis on the results presented, and not on whether the netcdf-4 format allows easy access to this type of data. Highlight the differences between using one method or another, for example.

Finally, although I do not consider myself an expert on this subject and I agreed to review this paper out of scientific interest since much of my work is using SSH fields and geostrophic velocities, so I consider that people like me should understand more about the processes you investigate but doing a paper where the methods are described so little it is complicated to understand what they do and what they are for. Therefore, I emphasize that you should make a better description of the methodologies, and even if you cite other works make a brief description that helps not to have to look for the cited work, that makes it easier to read and in itself that they read more of your work.

Round 2

Reviewer 2 Report

the authors have revised appropriately

Author Response

The paper has been significantly improved thanks to your feedback. 

Thank you for your time and involvement.

Best regards,

Ghita.

Reviewer 3 Report

Review of SARAL’s Full Mission Reprocessing: improvement with the GDR-F standard

I’d like to thank the authors for their responses and changes to the manuscript. While the paper is much approved, a few additional minor corrections are necessary to those revisions.

I continue to feel that the conclusion section continues to be weak. To address my previous comment, the authors only inserted a short statement about the improvement in the crossover variance from 5.5 cm to 5.2 cm. It is difficult to understand which improvements between GDR-F have had the most significant impact on this reduction. From the orbit error estimates for POE-F, it seems that roughly 2/3 of the reduction in SLA error is from the orbit error, and that only 0.1 cm improvement has come from the changes to the other corrections. Given that Ka-band range is more accurate than Ku-band range, it is surprising that GDR-D Jason data outperforms GDR-F SARAL data. As I previously commented, I feel that this paper’s conclusions are insufficient and could be addressed by including an error budget for GDR-F observations.

Line 43: I don’t think this is correct. The ‘F’ refers to the full set up revisions to the corrections, not just the POE-F orbits.

Line 112 NOA -> NOAA

Figure 15: The mean and std under the figure don’t appear to match the plot range. Are they in units different from cm^2?

Figure 17 caption: Bellow -> Below

Figure 18: Can you confirm the time step for the AltiKa GMSL? Is it one point for each cycle or every 10 days? The time step should be documented in the paper.

Reviewer 5 Report

Thank you very much for making the corrections and improving your work.

Author Response

(The authors gave the same response as above.)
